# Leveraging the Inductive Bias of Large Language Models for Abstract Textual Reasoning

**Christopher Michael Rytting**
Department of Computer Science
Brigham Young University
Provo, UT 84602
chrisrytting@byu.edu

**David Wingate**
Department of Computer Science
Brigham Young University
Provo, UT 84602
wingated@cs.byu.edu

## Abstract

Large natural language models (such as GPT-3 or T5) demonstrate impressive abilities across a range of general NLP tasks. Here, we show that the knowledge embedded in such models provides a useful inductive bias, not just on traditional NLP tasks, but also in the nontraditional task of training a symbolic reasoning engine. We observe that these engines learn quickly and generalize in a natural way that reflects human intuition. For example, training such a system to model block-stacking might naturally generalize to stacking other types of objects because of structure in the real world that has been partially captured by the language describing it. We study several abstract textual reasoning tasks, such as object manipulation and navigation, and demonstrate multiple types of generalization to novel scenarios and the symbols that comprise them. We also demonstrate the surprising utility of *compositional learning*, where a learner dedicated to mastering a complicated task gains an advantage by training on relevant simpler tasks instead of jumping straight to the complicated task.

## 1 Introduction

Natural language processing (NLP) has seen major progress thanks to probabilistic language models (LMs) like GPT-2 [1], BERT [2], and T5 [3]. These are pre-trained in a general, task-agnostic way on large corpora of unstructured text and then fine-tuned on specific tasks. This method has achieved state of the art performance on popular NLP tasks like question-answering, textual entailment, text summarization, neural machine translation, and more [4].

These models are not just impressive for the high scores they achieve on quantitative language tasks; the text they generate often reflects patterns of "real-world" structure, suggesting that embedded in the weights of the LM is implicit knowledge of physics, object persistence, containment, spatial relationships, causal mechanisms, material properties, and other common-sense knowledge central to human reasoning and intuition. If that is so, then they ought to provide a useful inductive bias in learning to perform symbolic reasoning tasks that mirror real-world tasks.

In this paper, we attempt to leverage and characterize this inductive bias by training reasoning engines that demonstrate some of the hallmarks of human reasoning ability: learning (a) rules (b) that generalize well (c) from few examples. Concretely, we fine-tune T5 on a suite of symbolic reasoning tasks and study generalization along multiple different axes beyond the normal test/train split: we examine cardinality generalization, object generalization, part-of-speech generalization, and show (in the spirit of curriculum learning) that LMs can leverage combinations of learned subskills to master complicated composite tasks with both better sample efficiency and higher terminal performance than learning directly on the complicated tasks.

35th Conference on Neural Information Processing Systems (NeurIPS 2021)

We see our contributions as four-fold. First, we demonstrate a high level of performance by connectionist models on tasks resembling symbolic classical AI tasks, demonstrating some symbolic ability of such models in light of recent calls to unify the principles of both symbolism and connectionism. Secondly, we demonstrate the breakdown of reasoning ability by manufacturing our own reasoning datasets that can be tweaked in systematic ways, instead of simply split into training/validation/test sets. This means that we can assess our models' ability to both interpolate and extrapolate in systematic, symbolic, and grammatical ways, and otherwise flex with changing distributions. Thirdly, we demonstrate the ability of large LMs to reason compositionally, that is, to learn two kinds of reasoning separately and then to combine those different kinds of reasoning on a novel composite task, to which they are both relevant. Lastly, we demonstrate the inductive bias we hypothesize is present in large LMs and can be leveraged to assist in the formation of reasoning engines.

## 2 Related Work

Transformer-based LMs [5] are the dominant LM architecture, including the original GPT models [1, 6], BERT [2, 3] and the recent 175 billion parameter GPT-3 [4]. They are trained via generic maximum likelihood learning on vast, unstructured text corpora, but often exhibit zero/few-shot learning abilities on a variety of NLP tasks, having grasped key mechanics of natural language by learning to simply predict a missing word; it is this inductive bias we hope to harness.

These models implicitly house a rich world model with concepts and relations. Petroni et al. query a language model (instead of a traditional, symbolic knowledge base) for relational data explicitly expressed in natural langauge [7]. Bosselut et al. expand this scope, attempting to generate explicit commonsense knowledge graphs using pre-trained LMs [8], and Bouraoui et al. perform relation extraction with BERT, which can be construed as finding and specifying edges [9]. This work treats the formation of explicit knowledge graphs using LMs, whereas ours treats the leveraging of implicit knowledge graphs using LMs.

Our tasks are inspired by classic examples of good old-fashioned AI, including Blocks World [10] and STRIPS planners [11], where a system's state, composed of symbols such as toy blocks, must be manipulated to achieve a desired end given a rule set. In contrast to these original problems, we *learn the symbolic rules* governing a similar system from examples. Our work is therefore similar in spirit to program induction, where a connectionist model must learn a structured program from examples; examples include the Neural Turing Machine [12] and the Differentiable Neural Computer [13]. Some work has attempted to teach a language model rules by explicit teaching through specialized datasets [14] or knowledge graphs [15]. We count on these rules being learned implicitly, as a side effect of the general learning objective of LMs. Weber et al. combine neural networks with logic programming to solve multi-hop reasoning tasks in natural language [16]. We don't insist on expressing explicit representations of symbolic rules learned, instead reasoning about rules learned by examining out-of-distribution performance.

Our tasks are similar to the bAbI dataset [17]. However, we focus on systematic study of specific dimensions of generalization that is impossible with bAbI. bAbI questions evaluate *whether* certain skills are possessed; our questions evaluate *to what extent* these skills are possessed, how they generalize, and when they break down.

Finally, the idea of compositional learning draws inspiration from multi-task learning, which was explored in [18] and more recently in a deep learning approach [19]. Supplementary Training on Intermediate Labeled-data Tasks (STILTs) was introduced in [20], which uses two stages of pre-training, where the first stage is unsupervised like with other pre-trained models, but where the second is on some data-rich intermediate supervised task.

## 3 Data Generation, Evaluation, and Training Protocol

The central questions we ask in this paper revolve around whether large LMs possess some world model or inductive bias that helps them to learn reasoning rules from few examples. These rules should be used to successfully extrapolate not just to novel instances, but to novel tasks. Upon learning to track objects across containers, for example, a language model should be able to generalize to new types of objects, marginally more complicated scenarios than it has seen before, and leverage skills already learned to progress more quickly than if it hadn't.

Table 1: Examples of scenarios for each task. Non-bold words indicate natural language templates that are filled in with randomly sampled words. Additionally, template phrases can repeat to describe a variable number of containers, rooms, objects, or navigation steps.

| Task | Prefix | Target |
|---|---|---|
| Container | *The **bin** contains a **ball** and a **snake**. The **box** contains a **quilt**. Took a **quilt** from the **box** and put it in the **bin**.* | *The bin contains a ball, a quilt, and a snake. The box contains no objects.* |
| Navigation Route | *The **garden** is to the **west** of the **kitchen**. The **bedroom** is to the **south** of the **kitchen**. To get from the **kitchen** to the **garden**, you must go* | *to the west.* |
| Navigation Result | *The **garden** is to the **west** of the **kitchen**. The **bedroom** is to the **south** of the **kitchen**. If you start in the **kitchen** and go to the **west**, you will end in the* | *garden.* |
| Hard Object / Composite Task | *The **kitchen** is to the **north** of the **garage**. The **garden** is to the **west** of the **kitchen**. The **bedroom** is to the **south** of the **garage**. There is a **bin** containing a **book** in the **bedroom**. There is a **box** in the **garden** containing no objects. Took a **book** from the **bin** in the **bedroom**. Went to the **north** twice, then went to the **west**. Placed it.* | *The box in the garden contains a book. The bin in the bedroom contains no objects* |

We show how LMs can generalize in these ways and others by systematically fine-tuning them on classes of scenarios where they can learn rules. Across tasks, our reasoners deduce the final state of an environment given natural language descriptions of an initial state and actions taken on it.

## 3.1 Task Types

**Containers.** In the first task, which we call *containers*, we manipulate objects in various containers and ask the reasoner to track the state of the environment. The initial state of the environment is a random allotment of *n_objects* objects into *n_containers* containers. The names of these objects and containers are sampled uniformly and without replacement from lists of candidates. Examples of such candidates are given in the appendix. Once sampled and organized, the objects and containers are converted into a plain English expression describing their organization. Then, there is a random manipulation of this initial state, where an object is randomly taken from a container and placed into another container. The task of the reasoning engine is to describe the final state of the environment.

For training, each scenario uses 2-8 objects and 2-3 containers, sampled uniformly. We construct sentences by sampling without replacement from a uniform distribution of container names and object names and filling a template accordingly. Our training set of object and container names comes from a proprietary linguistic dataset, which has data on commonness, part of speech, etc. of each word. We divide this dataset into nouns and verbs and subtract their intersection from both. We split the unique nouns into a train set (n=36566) and a validation set (n=12189) and a container set (n=9), the former two of which are to be used as object names and the latter of which is to be used for container names during training. For each set (train and val), we find the 2000 most common and join the dataset on concreteness ratings from Brysbaert et al. [21] to take the 2000 most concrete nouns from each set. We also generate a list of random strings by sampling uniformly from single digit integers and lowercase English characters, ranging in length randomly from 5 to 10.

**Navigation.** In both versions of navigation, a natural-language description of a map of the environment is provided to the reasoning engine. It is generated by sampling from a list of common locations including "kitchen", "garden", and others. These locations are then composed into a grid where they are in north-south-west-east orientation to each other. In the first task, called *navigation route*, the reasoner is additionally given a starting point and a destination. The reasoner must provide a valid route from origin to destination. In the second task, called *navigation result*, we require that the reasoner, given a starting point and a route, determine the location where they would find themselves in the map. For training, the maximum number of locations is 8 and the minimum is 3.

**Composite task.** We are interested in assessing the ability of a reasoning engine to perform a task which combines multiple elemental types of knowledge. Specifically, can a learner master two skills separately, and then leverage knowledge from both to perform the composite task better and more quickly than it would have by learning directly on the composite task? This task, called *hard object*,

is intended to be a composition of navigation and containers. The reasoner is given a verbal map of the world and an action taken. In this action, an object is taken from its container, carried on a route indicated by successive moves in cardinal directions, and placed. The reasoner must then describe the new state of the containers.

## 3.2 Generalization Types

Given our overarching interest in large LMs' ability to yield reasoners that generalize well, we craft several types of experiments to probe the bounds of different kinds of generalization, namely:

**Cardinality generalization.** This tests a model's structural understanding of the domain. If a reasoner is trained on scenarios with $k$ objects, steps of navigations, rooms, etc. can it generalize to more than $k$ objects, steps of navigations, rooms, etc.?

**Object generalization.** This is a semantic test of whether or not the reasoner can leverage prior knowledge of English to generalize to new, never-before-seen objects. For example, if several different container training scenarios are composed of objects from different distributions of words (e.g. 2000 concrete nouns vs. 2000 most common nouns vs. 2000 randomly sampled nouns), which distribution of training scenarios results in the best model generalization to scenarios composed with new, previously unseen nouns?

**Part-of-speech generalization.** This is another semantic test based on the idea that a reasoner that understands language will perform better on "right" words than on "wrong" words. For example, on the container task, the model should perform well when nouns are objects and poorly when verbs or random strings are. Intuitively, a scenario such as "The bin contains a dethrone and a transpose" makes less sense, and is statistically less likely, than a scenario such as "The bin contains a ball and a snake"; the naturalness of the scenario descriptions should positively correlate with generalization. We hypothesize that natural nouns will work best, since nouns are often sensible things to move from one container to the other, and that arbitrary strings and verbs will both decrease performance.On the other hand, random strings have probably never been encountered by the model, and it might learn to simply copy whatever tokens are in specific places, treating strings as opaque IDs, symbols without meaning. Verbs should actively confuse the model, since a verb's linguistic role is structurally and semantically different than a noun's; it is a symbol not without meaning, but with the wrong meaning.

**Reasonable phrasing generalization.** Finally, what if we replace the templates themselves? Instead of replacing the objects in a scenario, we replace the English scaffolding we insert them into by mapping deterministically from each English word to a gibberish word composed of English morphemes but possessing no meaning (See 4.5). A reasoner for which language is meaningful will have a harder time adjusting to this task, while a reasoner for which language is simply a sequence of meaningless strings will struggle no more with this task than with its original English version.

## 3.3 Base Model and Training Details

On all experiments, we use T5's 3 billion parameter architecture, either fine-tuning a pretrained model or training from scratch. We do so on Nvidia Tesla V100 32GB GPUs with a batch size of 1 and a learning rate of 0.003. We train a different reasoning engine for all three elemental tasks for 1000 total steps. After evaluating them on their respective tasks, we train them on the other base task for 1000 steps and eventually on the composite task for 1000 steps (3000 total steps).

## 3.4 Metrics

For each experiment, we gauge performance using three metrics. The first is **exact equality** of true final state. This is obviously the highest standard, as it mimics perfectly the reasoning we would expect a human to carry out after having learned the rules and format of the dynamical system. The second metric is **substring equality**. We want to see how many individual statements from the predicted final state are contained in the true final state. Sometimes the reasoning is flawed, but on target (e.g. if the reasoner predicts, after moving a hammer from a box to a bin, that it is in both the box and the bin, it should be given partial credit). The third metric is the standard **BLEU score**, to see how similar the sentences are at the individual word level.

**Interpolation and Extrapolation**. Throughout our experiments, we assess different kinds of generalization. The first, which we term *interpolation*, refers to testing a LM on new instances that are

drawn from the same distribution as the training set. For example, in the container task, we might test on scenarios that use already-seen objects and containers, but arranged in novel scenarios. The second, which we term *extrapolation*, refers to testing a LM on new instances that are drawn from a different distribution than the training distribution. For example, in the container task, we might test on scenarios that involve new, never-before-seen objects, containers, or cardinalities.

**Structural and Semantic Generalization.** Finally, we distinguish between *structural generalization*, where we test things like cardinality generalization, and *semantic generalization*, where we explore new words, or new types of words.

# 4 Results

We now systematically test interpolation and extrapolation, assessing both semantic and structural generalization. We begin with several baselines, and then explore increasingly difficult tasks.

## 4.1 Comparison with Baseline Methods

Since our claims have to do with the inductive bias of large, pre-trained LMs, we first establish a wide baseline gulf between pre-trained models and those trained from scratch (tabula rasa). We train two such versions of T5-3B on the same training set (1000 scenarios with maximums of 8 objects and 3 containers and minimums of 2 objects and 2 containers) and then measure two kinds of both interpolative and extrapolative performance: systematic, meaning the number of objects and containers, and seman-

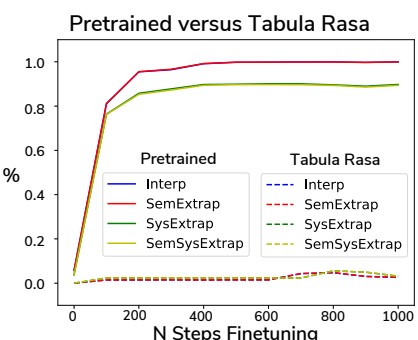

Figure 1: Model performance on 4 experiments over the course of fine-tuning. Compare solid lines (off-the-shelf pretrained models) and dotted lines (tabula rasa models) of the same color.

tic, meaning the words used for objects in reasoning scenarios (See 3.4 and 3.2 for more info). Thus, the two models' performance is compared on 4 total experiments with 7200 examples each: **Interp** - same number of containers and objects and same words used for object names (as training set); **SemExtrap** - same number of containers and objects (as training set), different words used for object names; **SysExtrap** - minimums of 4 containers and 10 objects and maximums of 5 containers and 19 objects, same words used for object names (as training set); **SemSysExtrap** - minimums of 4 containers and 10 objects and maximums of 5 containers and 19 objects, different words used for object names (than training set); The average BLEU scores for each model checkpoint (taken every 100 steps during fine-tuning/training) are displayed in Figure 1. These baselines make clear that pre-trained language models that have been fine-tuned wildly outperform language models trained from scratch.

## 4.2 Containers

We now begin our primary experiments, starting by testing cardinality generalization. We train on scenarios with a maximum of 8 objects and 3 containers, and validate structural generalization on scenarios with up to 19 objects and 5 containers. The results are shown in Figure 2. The model is able to correctly predict the exact string a majority of the time even well outside the training domain. This kind of generalization could only be reached by grasping, to some extent, rules like those used when enumerating long English lists. This is one way in which the inductive bias of LMs aids in the reasoner's good performance.

We now test object generalization and part-of-speech generalization. Here, we train models with scenarios generated by parameterizing our templates with various sets of nouns, and then testing by parameterizing with a different set of words. The results are shown in Table 2. Each row represents a different training set; columns represent performance on different test sets. We tested four different training sets: "all" represents nouns sampled uniformly from our set of all 36,566 nouns. "2k common" represents the 2000 most common nouns, "2k concrete" represents the 2000 most concrete nouns; "2k random" represents 2000 randomly sampled nouns. There are 3 groups of columns, representing different validation conditions. "Training words" represents a validation condition where the reasoner was tested on new scenarios that used the same set of nouns (eg, a different mix of

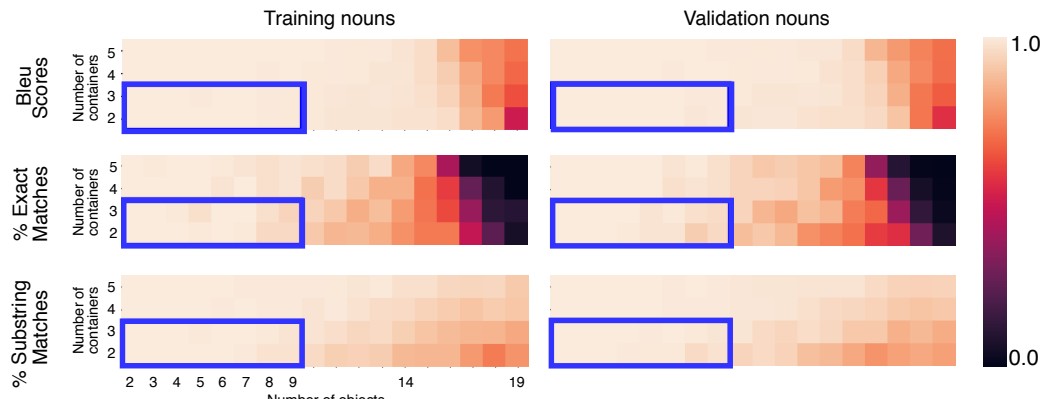

Figure 2: The blue box encompasses the training domain. Each cell is the average of performance over 100 generated instances for each combination of *n_objects* and *n_containers*. There is an obvious drop off higher in the number of objects, but this is a truncation issue due to long sentences (note that substring scores don't degrade nearly as much).

Table 2: Detailed results on the container task, showing various aspects of interpolation and extrapolation, both to new scenarios, new types of words, and new surrounding texts.

| | training words | | | | validation words | | | new pos | | |
|---|---|---|---|---|---|---|---|---|---|---|
| | all | concrete | common | 2000 | all | concrete | common | verbs | toverbs | random |
| bleu | | | | | | | | | | |
| all | 0.95 | 0.98 | 0.99 | 0.95 | 0.95 | 0.98 | 0.99 | 0.99 | 0.95 | 0.82 |
| 2k common | 0.93 | 0.96 | 0.95 | 0.93 | 0.93 | 0.96 | 0.96 | 0.99 | 0.95 | 0.81 |
| 2k concrete | 0.93 | 0.96 | 0.97 | 0.93 | 0.93 | 0.96 | 0.96 | 0.99 | 0.95 | 0.78 |
| 2k random | 0.91 | 0.94 | 0.95 | 0.91 | 0.91 | 0.94 | 0.95 | 0.99 | 0.95 | 0.81 |
| exact | | | | | | | | | | |
| all | 0.79 | 0.88 | 0.91 | 0.79 | 0.78 | 0.87 | 0.90 | 0.86 | 0.66 | 0.44 |
| 2k common | 0.60 | 0.71 | 0.65 | 0.59 | 0.60 | 0.69 | 0.67 | 0.83 | 0.63 | 0.37 |
| 2k concrete | 0.60 | 0.68 | 0.75 | 0.59 | 0.59 | 0.69 | 0.71 | 0.86 | 0.60 | 0.31 |
| 2k random | 0.56 | 0.61 | 0.68 | 0.55 | 0.55 | 0.60 | 0.64 | 0.87 | 0.63 | 0.40 |
| substring | | | | | | | | | | |
| all | 0.95 | 0.97 | 0.96 | 0.95 | 0.95 | 0.97 | 0.96 | 0.98 | 0.97 | 0.96 |
| 2k common | 0.82 | 0.90 | 0.85 | 0.82 | 0.83 | 0.88 | 0.87 | 0.97 | 0.95 | 0.93 |
| 2k concrete | 0.83 | 0.87 | 0.91 | 0.82 | 0.83 | 0.88 | 0.89 | 0.98 | 0.92 | 0.84 |
| 2k random | 0.80 | 0.83 | 0.86 | 0.80 | 0.80 | 0.83 | 0.85 | 0.98 | 0.94 | 0.91 |

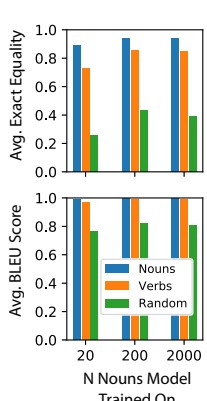

Figure 3: Models trained on concrete nouns were most confused by verbs and random strings.

containers and objects, or a different number of them). "Validation words" represents a condition with new scenarios using never-before-seen words of a specific type. "New POS" represents words that are an entirely different part of speech, or random words. The general conclusion is that, regardless of the metric used, it is best to train on the largest non-specific set of nouns possible. Doing so gives good generalization across a wide variety of alternative words, including verbs and random strings.

We now explore the distributions of the training nouns. Many of the words passed between containers were nouns that would not be passed in between containers, like "year", and so we conducted an experiment where we trained on the 2000 most concrete nouns, a random subsample of 200 of those nouns, and 20 of the most sensible nouns to be moved between containers, like "marble" or "mouse". As can be seen in Figure 3, the model trained on sensible nouns gets most confused at verbs and random strings, relative to its peers. This suggests that the distribution of very concrete nouns is farther from the distribution of verbs than the larger distribution of concrete nouns.

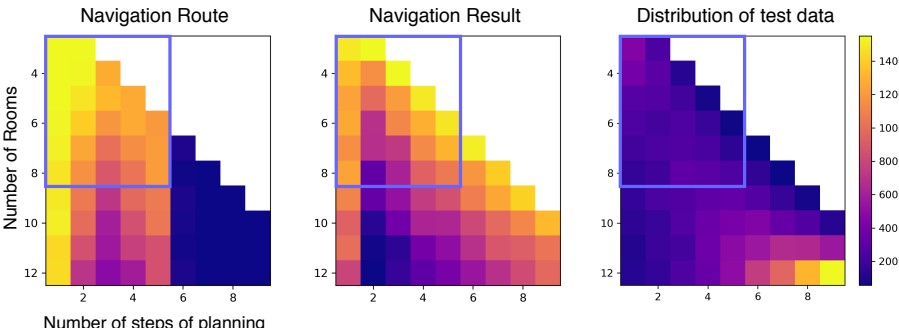

Figure 4: Validation performance on the navigation route and navigation result tasks. Entries represent exact string equality. Training regime is outlined in blue box. See text for details.

## 4.3 Navigation

We now turn our attention to the navigation and navigation route tasks. In this section, we use exact string accuracy as our metric, as both substring and BLEU scores ignore the (critical) sequential nature of the predictions.

### 4.3.1 Interpolation and Extrapolation

Our primary results are in Figure 4. For both the navigation route and navigation result tasks, we trained on distributions of maps that contained between 3 to 8 total rooms; this resulted in total path plans that were 1 to 5 steps long.

In the middle, we see results for the navigation result task (where the model must predict the result of a specific sequence of actions). Here, we see several interesting phenomena. Overall, the model does a good job of making accurate predictions, with a 79% marginal accuracy over the training regime. There are two situations where the model performs especially well: the first column represents tasks where there is only a single step in the route; the model shows a strong ability to extrapolate to any number of rooms. The second case is the "diagonal" in the upper-right. This represents situations where the map is a linear chain, with very few junctions in the map. In this case, the model also shows a strong ability to generalize to new situations involving a new number of rooms, and a new number of steps. It is unclear why the model has such trouble with two-step planning problems, although we hypothesize it may have something to do with the training distribution, as discussed later.

In contrast, on the left of Figure 4, we see results for the navigation route task. Here, we see similarly strong performance in the training regime, with a 82% marginal accuracy. However, extrapolating to new situations shows mixed results: the model is easily able to extrapolate to any number of rooms, and like the navigation result task, it performs especially well in the case of single-step planning. However, it seems to be completely unable to generalize to new numbers of steps needed in the planning process.

What accounts for the difference? The biggest difference between the navigation route and navigation result tasks is the format of the output: in the navigation result task, the model always outputs a single word (the room name), regardless of the complexity of the map or the plan. In contrast, on the navigation route task, longer plans require a longer output. The model seems to struggle with having to generate sentences that are longer than any it has seen before. While we have used exact string equality in all of the tests reported here, we note that on longer routes, sometimes the model's output would be a substring of the correct output, as in the example below, which is correct, but missing the final step:

```
    Target:  to the west, then to the west, then to the north, then to the north, then to the north, then to the north
Prediction:  to the west, then to the west, then to the north, then to the north, then to the north
```

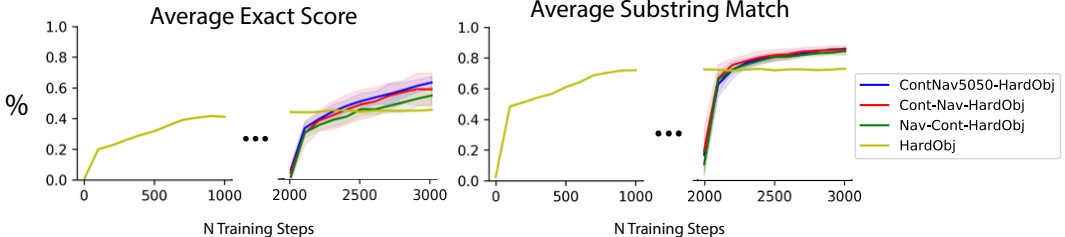

Figure 5: Confidence bands represent one standard deviation above and below mean performance under each metric. The composite learners exhibit better sample efficiency and a higher absolute performance ceiling than the learner dedicated exclusively to the composite task.

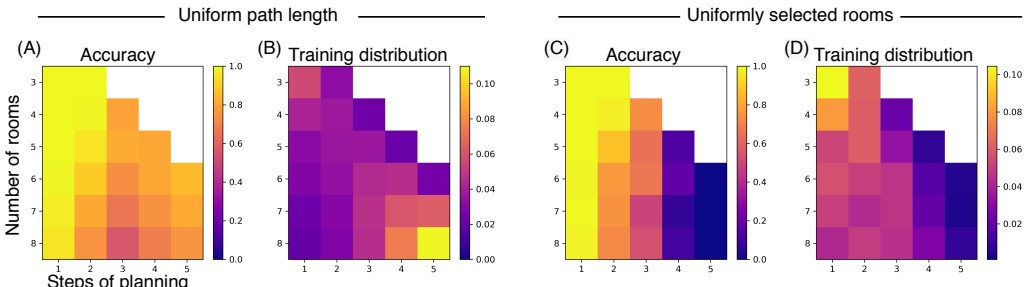

Figure 6: Extrapolation ability as a function of training set distribution. See text for discussion.

## 4.4 Compositional Reasoning on the Composite Task

Finally, we tested the *hard object* task, where the reasoner must consider objects, containers, and maps. To explore this, we test four conditions. We train a learner exclusively on the task (HardObj). We also train a learner first on the navigation task for 1000 steps, then on the container task for 1000 steps, then on the hard object task for 1000 steps (Nav-Cont-HardObj). We also reverse the order of navigation and container pre-training (Cont-Nav-HardObj), and we test a random mixture of pretraining, with 2000 steps of a 50/50 mixture of sentences drawn from both tasks (ContNav5050-HardObj). For each regime, we trained 10 models and averaged their performance on a test set of 5000 held-out examples.

The results in Figure 5 are surprising: the models that learned first on other kinds of tasks learned the composite task more quickly and achieved better ultimate performance. Like curriculum learning, this seems to indicate subskills can persist and be ported to superskills for improved sample efficiency. Instead of fine-tuning on a single hard task (and thereby gaining proficiency in only that task), a better strategy may be to fine-tune on a wide variety of elemental tasks, and focus more on combining them, thereby potentially solving not just one, but a combinatorially large number of complex tasks.

### 4.4.1 Importance of Training Set Distribution

The T5 model seems unusually sensitive to the training set distribution. Figure 6 explores this. In our first attempt at creating a training distribution for the navigation tasks, we randomly created a map with a uniformly selected number of rooms, and then randomly selected source and destination rooms. This resulted in the performance shown in the right-hand panels (C and D) of Figure 6. Panel (D) shows most of the training data is concentrated in the upper-left corner, meaning short paths in small maps. As a result (panel (C)), the model was almost completely unable to model length 4 or length 5 plans, because they rarely appeared in the training set.

An alternative is to sample a desired path length uniformly, and then generate a map, source, and destination with that length. This results in the training set distribution shown in panel (B), which concentrates many more examples on longer paths in larger maps, but with fewer examples of two-step plans. The resulting accuracy is shown in Panel (A). Here, we see that accuracy is greatly improved for four- and five-step plans, although two-step accuracy suffers. However, even the

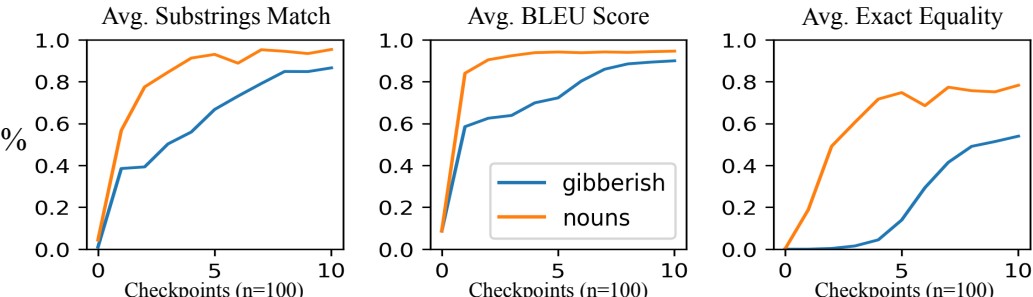

Figure 7: The advantage of English scenarios over equally consistent reasoning scenarios in an invented language is evidence for the aid of language models' inductive bias.

carefully constructed training distribution with uniformly sampled path lengths does not induce generalization to paths longer than 5 steps, as shown in Figure 4. Understanding this phenomenon is an important direction for future research.

### 4.5 Does all that reading really help?

At the heart of our paper is the idea that natural language provides a useful inductive bias. To explore this directly, we modify our object-container task in the following way: We devise a one-to-one mapping from each word in our sensible English templates to words in an invented gibberish. Since the mapping is one-to-one, the English templates and the gibberish templates are identically structured but with substituted words. For example, the word "The" is mapped to the gibberish word "Xrq", the word "contains" to "sixnqkxb", etc. Noun slots in both templates are still filled with the original English nouns (i.e. not substituted with gibberish). Figure 7 shows the results: it is harder for T5 to master the gibberish domain. If there were no inductive bias of English-trained models aiding our learners on English tasks, then the mastery of these two grammars would grow at the same pace, since they are just as structured as one another and thus presumably as predictable and learnable as one another. This is a bit of evidence in favor of the notion that the inductive bias of large LMs does, in fact, aid in learning quickly and generalize well in several different senses.

## 5 Ethical and Risk Considerations

Since this paper studies what are essentially toy tasks, we don't consider the risks for our methods to be particularly high. However, we believe in the eventual application of these models in real-world domains (e.g. robotics and planning), and this work contributes to a needful preliminary understanding of these models' behavior, since they are difficult to control and predict. We urge applications of these models to take into account this general risk along with work such as ours which helps to characterize it. It is dangerous to use language models in high-risk domains (consider a surgical robot or a planning agent for defusing explosives) without extensive understanding of their interpolative and extrapolative generalization.

## 6 Conclusions and Future Work

The central goal of this paper has been to investigate if connectionist LMs can learn something akin to symbolic rules that generalize in "natural" ways. We have shown strong performance on a suite of reasoning tasks, including tasks that we consider to be simple–an object manipulation task and a navigation task–and more difficult–a composite task that combines the other two. On the container task, we show near perfect performance on the training domain and considerably far outside of it and an eventual marked drop-off in performance. On the navigation task, we show differing patterns of generalization on different subtasks, with slightly lower performance than on the container task. We have also demonstrated some of the boundaries of generalization by identifying several different kinds of generalization and measuring them. We do not claim to have explored all the axes of generalization possible in the study of these language models, but rather shown how the generalization ability of

LMs can be probed. But these results are exciting because the models perform so well outside of the training domain that it seems as if some general, symbolic rules have been learned.

Beyond generalizing just to new examples on learned tasks, we have demonstrated that these models are aided in the performance of complex tasks by learning first on elemental tasks. This hints at the prospect of learning difficult tasks more quickly and better by learning more simple tasks first.

We have additionally argued that natural language provides a powerful inductive bias for symbolic tasks by comparing on scenarios across different, but equivalent grammars, one in natural language and one in an invented language. Intuitively, learning the distribution of language that describes the world helps a learner to understand the distribution of that world itself. Thus, it is plausible that LMs might be effective ways to provide autonomous agents with priors and world models.

Implicitly, we have demonstrated that connectionist architectures are capable of strong performance on some classically symbolic tasks. This is done in light of recent calls to incorporate guiding principles of symbolic AI into connectionist models; it is exciting that language models seem to possess at least some of the rule-learning and generalization that humans possess, as opposed to the mere ability to recognize patterns and interpolate over a well-explored training domain.

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

## References

