# OpenReview forum: "Leveraging the Inductive Bias of Large Language Models for Abstract Textual Reasoning"
_NeurIPS.cc/2021/Conference — NeurIPS 2021 Poster_

### Official Review · Reviewer_hX8A · 2021-07-10

**Rating:** 6
**Confidence:** 4

**Summary:**

This paper aims to explore whether the inductive bias of pre-trained language models can support symbolic reasoning tasks. Three different types of tasks are designed for the above objective, including a container-based task, a navigation task and a composite task. In order to verify the generalization ability of the model, the paper also designed different kinds of generalization for probing, including cardinality generalization, object generalization, POS generalization and reasonable phrasing generalization. T5 is fine-tuned and evaluated on these tasks, with some interesting findings observed, which I think are the key contribution of this work: (1) for container-based task, T5 shows good generalization and prediction capability; (2) for navigation task, T5 can also do a good job but when the number of inference steps changes, its performance will drop. (3) for composite task, T5 can do a good job  by learning in a curriculum learning way.

**Limitations And Societal Impact:**

no negative societal impact

**Main Review:**

The objective of the paper is clearly clarified. The writing is easy to follow. The findings in the experiments are interesting. It is a good paper to study how well pre-trained LM can perform on reasoning-required tasks and its generalization capabilities on different aspects. I appreciate this work but think the datasets built within this paper is limited to specific domains and scenarios. It will be better if the paper can cover results on other open domain datasets as well, such as commonsense QA, mathematical reasoning, etc.

**Time Spent Reviewing:**

2.5 hours

---

> ### Author Response · Authors · 2021-08-10
> **Response to Reviewer hX8A**
>
> We thank the reviewer for their thoughts. Indeed, the datasets are limited to specific domains and scenarios. We thought it important to very carefully and thoroughly explore the dimensions of generalization that we did. For this small battery of tasks and with the detail included for each task, we couldn't include more tasks for lack of page space. With the publication of this paper, though, we are in position to write many more papers examining those domains and scenarios, expanding the field's understanding of what kinds of knowledge language models do and do not possess. The reviewer's suggestions of commonsense QA, mathematical reasoning, etc. are excellent places to start.

---

### Official Review · Reviewer_VwQ8 · 2021-07-13

**Rating:** 7
**Confidence:** 5

**Summary:**

In this work, the authors investigate if large scale pre-trained language models (such as T5) can provide inductive biases that are useful for solving language-based symbolic reasoning tasks, and furthermore, generalize to unseen settings.

First, the authors define four types of tasks. While all the four tasks are formulated as sequence generation given prompts (prefixes), they require different kinds of reasoning:

1. Container. The prefix text describes an initial states of the world and a sequence of transitions, the target sequence needs to describe the ending states.

2. Navigation Route. The prefix text describes a map as well as two positions in the map, the target sequence is the navigation path from one position to the other.

3. Navigation Result. The prefix text describes a map, a starting position, and a navigation path, the target is the ending position.

4. Composite Task. A mixture of container and navigation tasks.

Then, the authors define a set of experimental settings targeting the generalizability of models, from a variety of aspects such as map sizes, object/container numbers, reasoning steps required as well as some linguistic properties (e.g., replacing nouns with words with other POS tags, or even made-up words).

The authors provide plenty of experiments, suggesting the pre-trained language models indeed have capability of learning and to some extent generalizing symbolic rules via language-based tasks.


**Limitations And Societal Impact:**

Since the current version of the paper uses a set of synthetically generated toy tasks, I do not see immediate negative societal impacts. However, because the work relies heavily on large scale pre-trained language models, which training data is difficult to control. The author should definitely discuss the potential negative effect or harm these techniques may bring in the revision.

**Main Review:**

## Synthetically generated data

Just to be transparent. I was one of the reviewers of this paper at ICML 2021. The authors have described in the "submission history" field about lacking of baselines. In addition to that, there were some ICML reviewers devalued this work because the used tasks are synthetically generated (and thus not natural language).

I want to emphasize my point that **synthetically generated tasks are equally valuable**.

I understand and agree that the ultimate goal is to model and perform textual reasoning on real world data. However, before stepping into natural language data, I believe it is meaningful to have synthetic environments that are controllable, helping researchers to better understand the problem. Although the tasks introduced in this work use templated language, it is clearly shown in the figures that they are not oversimplified (there are areas in the heatmaps where the models fail to generalize). It might make less sense to use natural language data when strong pre-trained LMs such as T5 still have trouble solving/generalizing on toy-ish tasks. Because otherwise, it might be difficult to analyze, for example, which part of the task is the model struggling on. Furthermore, there are many examples in the NLP communities that use synthetic datasets (templated language) as a starting point of certain research directions, such as bAbI (Weston et al., 2016), TextWorld (Côté et al., 2018) and many more.

One potentially interesting thing to try might be, back-translating (e.g., EN -> DE -> EN) the teamplated descriptions, and investigate to what degree the language models generalize on this dimension of "naturalness".

## Overall comments

This work provides a set of well-designed experiments with convincing results that explore an interesting and important direction.

Since the flourishing of the large scale pre-trained language models, SOTA scores of neural models on various NLP tasks have increased by a lot. On some tasks, some researchers believe neural models have achieved human-level performance, while others argue that neural models may be somehow exploiting biases and trivial cues injected into the data or model unconsciously, rather than really doing reasoning.

To that end, the community do require works that help us understanding what and why such language models can learn, and to generalize. This work falls into this category.

Among the experiment designs and presentations, I especially like the heatmaps, which clearly show how the model gradually loses its generalizability when certain dimension of task becomes more difficult.


## Typos and minor things

1. L217: There are 3 groups of columns

## References

1. Towards AI-Complete Question Answering: A Set of Prerequisite Toy Tasks. Weston et al., 2016.

2. TextWorld: A Learning Environment for Text-based Games. Côté et al., 2018.

**Time Spent Reviewing:**

5

---

> ### Author Response · Authors · 2021-08-10
> **Response to Reviewer VwQ8**
>
> We thank the reviewer for their support of this submission both in the ICML and NeurIPS review processes. We will be sure to fix the typo in L217 and to discuss potential negative/harmful effects that may come downstream of these techniques in the camera-ready version of the paper.

---

### Official Review · Reviewer_gnNv · 2021-07-17

**Rating:** 3
**Confidence:** 4

**Summary:**

This paper investigates whether pre-trained language models (T5 in this case) learn priors which support symbolic reasoning tasks in the process of optimizing their purely linguistic objective.  The performance of fine-tuned pre-trained LMs is evaluated on two symbolic reasoning tasks -- object location across a number of boxes, and a room navigation task -- and compared with identical models trained from scratch.  The results touch on a number of topics, but generally show (1) good reasoning accuracy when starting from the pre-trained t5 model, (2) abysmal performance by the from-scratch models, (3) a good degree amount of generalization performance for t5 to reason beyond the fine-tune scenario, and (4) that learning subtasks improves performance on a compositional task.


**Ethical Concerns:**

No.

**Limitations And Societal Impact:**

Yes.

**Main Review:**

There's a lot to like in this paper.  The presentation and writing quality overall is just fantastic.  Not only a clear presentation, but a great style that makes the reading really enjoyable.  The authors have also been very thorough in listing the considerations behind each experiment, and approached each from multiple angles to get rid of confounds.  It's clear that a lot of thought lies behind each one.

My main criticism, and a crucial one, is that the experiments don't go far enough to show what the authors purport to show.  This boils down to a few things:

1) Unclear separation between language understanding and world knowledge.

The story laid out in the introduction, especially paragraph two, is one of text mirroring patterns in the real world, and the potetial for LMs to encapsulate this knowledge.  The authors provide a long list of potential types of world knowledge, and whether LMs capture any of these is very exciting and a very timely research question.  Just to underscore that this sort of information is central to the paper's argument, L25, "-If that is so-, then they ought to provide a useful inductive bias in learning to perform symbolic reasoning tasks that mirror real-world tasks".  I'll call this "world knowledge" for the remainder of the review.  Those statements really set the stage for the tone of what this paper is claiming to show, it follows from the reasoning in L7-8 of the abstract, etc.  Suffice to say, from that point I would expect that the paper ends by conclusively showing that this world knowledge is there in their chosen pre-trained LM.  And I think this is where it fails.

Desired interpretation of results: Container task.  Pre-trained model performs vastly better on extrapolation and overall when compared to from-scratch model, because it has embodied world knowledge, (maybe object persistence, containment, from the introduction list).  It shows good generalization ability because it T5 has induced reusable structures of reasoning / world knowledge.  It performs better with concrete nouns, then nouns, then verbs, because utilizing these objects in this manner does not make sense.

Alternate interpretation of results: Container task.  Pre-trained model performs vastly better when compared to from-scratch model because it has some rough representation of the word classes of these words.  It generalizes better to new words because it has similar representations of those words.  It shows good generalization ability because it can do so on top of well defined word class representations (completely apart from their role in reasoning).  It performs worse on verbs because it has trouble refining representations of ungrammatical sentences, and reasoning on top of non-sensical sentences suffer.

In the first interpretation, I'm really interested to further understand these induced world knowledge patterns more and I really want to accept this paper.  In the second interpretation, it's just shown that pre-trained LMs have really good representations of basic linguistic categories, and that fine-tuning for all sorts of tasks is hugely effective, something the community is already quite aware of.  It's still interesting to see applied to these bAbI-like tasks, but then again, that's a different sort of paper.  In that setting, I would have liked to have seen a comparison with those models and more practical advantages/disadvantages/failure cases described.
And to the second interpretation, where basic linguistic knowledge accounts for everything,  I don't see any evidence here that would rule this out.  That's not to say that the hypothesis is wrong -- indeed, I would suspect there is a lot of world knowledge in there -- but I don't believe the authors have shown that conclusively.  I think the tasks chosen here make it difficult to really talk about real world knowledge.

The authors sometimes waffle between these two.  For instance, at the end of section 4.5, the English->gibberish conversion, the authors state that their experiment provides "evidence in favor of the notion that the inductive bias of large LMs does, in fact, aid in learning quickly and generalize well in several cases".  In this instances, the story is muddled, because we know that a general inductive bias in large LMs helps learning in many areas, it's basically the state of NLP at the moment, and so it needs no supporting evidence.  It's whether the inductive bias is based in real world knowledge that is unique to this work and pushed for in the introduction, and these experiments do not separate whether the advantage of the English model comes from linguistic knowledge or world knowledge.

It actually strikes me perhaps as evidence of point (3) below, where the gap between this lines correlates roughly to the combined linguistic and world knowledge the LM learns, while the reasonably high performance of the gibberish model is basically a testament to these training procedures getting the weights in a good position for transfer to unrelated tasks.

Another possibility of more concretely discovering evidence of world knowledge would be probing the representations in a targetted way.  Combining that with the sorts of experiments provided could strengthen the overall impression of the work.

2) Poor Baselines

In terms of experimental setup, of course it is tough to get at exactly what types of practical knowledge / inductive bias (apart from linguistic knowledge) these models contain, but I think some problem stem from the chosen baselines.  The authors use the same T5 model for the pre-trained model and for the from-scratch model.  This was a terrible idea in my opinion.  No reader should accept a 3 billion parameter model trained for 1000 steps to be a suitable baseline for comparison -- it is clear that this model was not given any chance to succeed.  I'm curious how smaller-parameter variants fair on the from-scratch setting.  I think it's quite clear that 3B parameters is not required for these sorts of tasks.

Speaking of similar tasks, the tasks are admittedly close to the FB bAbI in spirit, however, going back to that original work, there was not so much difficulty in achieving good results with models that we would considering quite basic by today's standards.  Why?  Presumably baseline model size and over-capacity.  But also thinking more generally, there were previously good results on bAbI tasks with simple memory-augmented models on top of word embeddings.  Is it similarly sound to claim that word embeddings had strong inductive biases that mirrored the world outside of their linguistic roles?  Of course it's also a question of training size and setting the tasks up to test generalization, but the rationale seems similar to the argument of this work.

I would have also liked to have seen some other models which are maybe capable of inducing linguistic knowledge relevant for the task, without any serious possibility of learning things like physics or spatial relationships, other examples of these properties, etc.

3.) Large pre-trained LMs can improve performance on unrelated tasks

An important point of comparison is the "Pretrained Transformers as Universal Computation Engines"[1], which was not mentioned in this paper, though it appeared on arxiv only a few weeks prior to the submission deadline.  Inn summary, the authors showed that pre-training on large text corpora provide a significant learning advantage when applied to completely unrelated domains (via fine-tuning).

When considering this submission in light of this finding, it raises the question of whether the conclusions in this work are true / whether pre-trained LMs have induced symbolic reasoning knowledge, or merely a result of more universal organization properties in the network.  It would perhaps be interesting to compare these results with models pre-trained on an equally large section of another language and test in English, especially if preposition-style relationships were expressed in different ways (since they seem fundamental to both tasks).

Regardless of the answer to these questions, this is an important point to discuss.

[1]
@article{DBLP:journals/corr/abs-2103-05247,
  author    = {Kevin Lu and
               Aditya Grover and
               Pieter Abbeel and
               Igor Mordatch},
  title     = {Pretrained Transformers as Universal Computation Engines},
  journal   = {CoRR},
  volume    = {abs/2103.05247},
  year      = {2021},
  url       = {https://arxiv.org/abs/2103.05247},
  archivePrefix = {arXiv},
  eprint    = {2103.05247},
  timestamp = {Mon, 15 Mar 2021 17:30:55 +0100},
  biburl    = {https://dblp.org/rec/journals/corr/abs-2103-05247.bib},
  bibsource = {dblp computer science bibliography, https://dblp.org}
}






Questions:

Does the model size matter?  I'm curious how small a T5 model has to be before it has significant effects on the reasoning task, especially if additional fine-tuning layers were added (of equal size for all T5 variants), and thus isolate the task-specific weights from the "induced world knowledge" weights.

The paper might benefit from more specificity in what the world knowledge the authors feel is relevant here.  For each, they could also describe whether previous NLP work has shown these properties exist in pre-trained LMs or in their necessity in other toy tasks.
And for instance, to choose one from the introduction that may be relevant, "material properties"?  For whatever property the authors think relates to their chosen experiments, could they discuss how it is necessary for the task?

L153: for fine-tuning, is this purely using the T5 weights or are any additional layers/heads added to the model?
L155: what constitutes a step?  A single training example?



Presentation:

Personally I find the use of "symbolic reasoning engine" to describe a fine-tuned T5 model a failed attempt at inflating the sophistication of the method.  At the very least, it prevents readers from more immediately latching on to the specifics of the model and therefore the study.

Fig 1., very difficult to spot the various overlayed lines, especially in a printed version.

In L19-20, The examples of tasks where pre-trained LMs achieve SoTA accuracy do not require patterns of ``real-world'' structure described in L22.  Where is the citation or evidence that generated text reflects the latter forms of knowledge?

L65, BLEU score is used to see how similar sentences are on subword levels?  But BLEU is a word-level metric?

L92, how are selected objects arranged into sentences?  What kind of diversity is there in these generated sentences or templates?

L102, most common... nouns / and join the dataset -- this description was a bit clunky


**Time Spent Reviewing:**

8

---

> ### Author Response · Authors · 2021-08-10
> **Response to Reviewer gnNv**
>
> We thank the reviewer for their incredibly thoughtful and thorough review. The concerns outlined are well-received.
>
> In response to criticism 1, about the difference between language understanding and world knowledge: The reviewer offers two possible interpretations of the paper, and in the review they are framed as alternatives. Surely, however, both of these interpretations are true to some extent. Raw natural language contains both "real world" and linguistic information. The language model's task is to accurately model that language, and to do so it must gain some understanding of both, at least to the extent that they both aid in predicting tokens in the target text. It is incredibly exciting that models can harvest both of these kinds of knowledge from reading raw text. In this work, we are not concerned with this philosophical difference between "real world" and linguistic knowledge, as we consider them to be tightly related. As a concrete example, the basic linguistic properties that the reviewer worries are the extent of what the model is learning have a great deal to do with the real world entities being described in that language (a word's part-of-speech determines whether the concept it represents is an object, and action, a characteristic, etc.). We also explore features that are not, at least in any traditional sense, basic linguistic properties, such as commonness or concreteness of a word. These features also speak to the nature of the concepts represented by words, and they are clearly learned to some extent by the model (consider the similarity between each group's training and validation scores in Table 2).
> This is not to say that these two types of knowledge are the same; the differences between them is surely significant and of great interest to us in the long run, but this paper was not intended to define that boundary or make claims about which kind of knowledge we are learning. Instead, we aim to investigate the generalization abilities of large language models given whatever they are learning in pre-training. We consider this to be a valuable technical contribution to the understanding of language models and their interpolative/extrapolative performance. There is very little mention in the paper of the real world knowledge possessed by language models, but in the camera-ready version, we are happy to adjust that language to be more measured and more exactly descriptive of the kinds of knowledge we measure in language models.
>
> In response to criticism 2, concerning our baselines: Every time we have submitted this paper, we have been penalized by reviewers for not including this baseline. We resisted including it because the types of baselines requested were either trivial or distracted from the point of the paper.
>
> Since we were studying the task of symbolic reasoning in the domain of natural language, any baseline had to be with another language model. We considered it trivial to show yet again that a transformer beat an LSTM or some other sequential model, a finding the field has replicated time and time again across many language tasks. We also didn't want to compare a pre-trained language model with a from-scratch language model for the reasons you mentioned. However, at least this latter study, although totally unsurprising, was closer to the points made in the paper, that a large language model's inductive bias can aid in generalizing on symbolic reasoning tasks in the domain of natural language and that this isn't a trivial task that can be learned just as quickly from scratch.
>
> With respect to your question about whether we would also argue that word embeddings have strong inductive biases, we would make no claim either way because we would defer judgment until there was empirical evidence one way or another. That is a very different kind of model, and comparing the two methods is a different paper.
>
> The third point about "Pretrained Transformers as Universal Computation Engines" is a fascinating one, and one we would love to pursue in future work. The experiments proposed by the reviewer would be important pieces of evidence in ascertaining the extent to which the network is a reasoner or an organizer, and what the differences/similarities might be between these two frameworks. We will be happy to include a bit of discussion referencing this recent work in the camera-ready version of the paper.
>
> As for the reviewer's specific questions:
>
> We did not experiment with smaller model sizes. Our large model didn't master the task, so it's likely a smaller model would have worse performance (by how much we can't say). Due to space constraints, we ultimately filled our paper with experiments focused on these generalization axes of higher interest.
>
> Our fine-tuned models come from the pure T5 weights; no additional layers/heads are used.
>
> We used a batch size of 1, so a training step constitutes a single training example.
>
> Selected objects are arranged into sentences by using the templates in Table 1.
>
> We would be happy to more clearly differentiate the overlaid lines in Fig. 1, fix the typo in L65, and clarify our description in L102.

---

> > ### Comment · Reviewer_gnNv · 2021-08-15
> > **Other baselines are necessary to distinguish the inductive biases of large language models vs. existing work**
> >
> > Thank you for clarifying these points.
> >
> > Skipping ahead, regarding the included baselines (T5 from scratch), I can certainly sympathize with the never-ending "Whac-a-mole"-style challenge of addressing every reviewer's concerns across multiple submissions.  As I'll discuss later, I (obviously) do think it's important to include reasonable baselines and I can see how without these T5-from-scratch comparisons it would come across as a glaring omission.  But I don't think we learn much at all from including these particular baselines either.  But now with a better understanding of why they are there, I think the question of their value can be put aside.
> >
> > Regarding criticism 1, you're quite right to say I may have jumped the gun on trying to distinguish "real world" knowledge from linguistic knowledge and focusing primarily on the former.  I think the introduction does position the reader to be thinking along those lines, but it's also a lot of assumption on my part.
> >
> > But I think it's also a set of assumptions I was quick to adopt because I'm searching for a story that I think is compelling enough to warrant acceptance, and the more "real world" knowledge that comes through to improve task performance, the more these results are going to read as novel in comparison to (1) older embedding representations, such as word2vec, fasttext, etc., and (2) other T5/large pre-trained (often transformer-based) LM work that probes these models for syntactic or word class information.
> >
> > I don't want to be too repetitive to the original review, but I think it's implied in making "the Inductive Bias of Large Language Models" the focus of the paper, that we are to some extent interested in the inductive biases of these models that are not also present in the previous "versions" of these models which we have been using for the past ten years.  I think if we cared about the task itself, the performance figures in the paper would be compelling in and of itself.  But since these are toy-ish tasks, I believe the focus must shift to understanding the inductive biases of these models with respect to existing literature.
> >
> > So I would really like to see evidence that these models offer something novel and significant, as it pertains to this task, over existing embedding models.  While I suppose it could be argued that this is a comparison for another paper, it's also true of almost any paper that the method should be contrasted with existing comparable methods in the literature, and the paper is currently lacking in this area.  What I wouldn't want to see is follow-up research using the same tasks but say, typical word embeddings, and show that they perform comparably well on this task.  And therefore show that really it was basic linguistic knowledge of word classes / object clusters (of the sort we had prior to the development of "Large Language Models") that were being tested in these experiments.  We know that representations learned by language models, simple skip-gram or large pre-trained ones, are useful to tasks that need this sort of information.  I don't see the value of showing this lesson once again, using situated game worlds in place of other typical NLP tasks.
> >
> > So it's unfortunate to me that intuitively the tasks seem to be designed to test this type of information (word class / semantic clusters) rather than some less typical knowledge.  Concreteness and commoness are not types of linguistic knowledge that I have ever seen studied in a representation probing paper, but I have seen T-SNE plots of word2vec-style embeddings that look clustered in ways that seem to reflect concreteness.  But I would really be swayed if the game worlds relied on information not found in existing non-contextual word embeddings.  Showing concreteness is one of these would be a step in that direction, but designing experiments to show more "understand[ing of] the distribution of that world itself" would be a big improvement.
> >
> > A possible additional reference: what you're trying to show swapping verb/noun representations reminds me a bit of the following paper:
> >
> > Investigating Human Priors for Playing Video Games
> > https://arxiv.org/pdf/1802.10217.pdf

---

> > > ### Author Response · Authors · 2021-08-22
> > > **Consider the dimensions we probe besides those in which existing methods excel; baselines can be improved to better distinguish large LMs and existing methods**
> > >
> > > Thank you for engaging so much in this review process. It's incredibly helpful to talk through all these issues to inform the direction of this work. And you're making some great points.
> > >
> > > We think you're right that the baselines you've suggested would tell a better story and certainly would speak to the specific hypotheses you float. On the other hand, we continue to think that the types of linguistic knowledge well-understood to be grasped by older models like embedding representations fall short of what we've demonstrated here. The axes of generalization we explore are useful not just to see whether the model grasps the task by substituting similar words (which we show with part-of-speech generalization, and could be an ability also possessed by embeddings), but also to understand the limits of cardinality generalization in terms of number of objects, containers, steps, and maps tracked, and this is a type of reasoning ability well-detached from learning word-classes. Moreover, it seems like your worry about the novelty of these abilities concerns our tests of part-of-speech generalization more than our tests of object generalization, since the categories we identify and test are more subtle than whether something is a verb or a noun. Of course, it would be really interesting to see whether word embeddings possess any of these capacities, but we thought and think that there is a lot more reason to suspect that large language models would  possess them than that word embeddings would, and for that reason we chose this class of models to base our paper on. We're not aware of any work showing word embeddings to be capable of the things we show in this paper.
> > >
> > > That being said, we believe the paper would be stronger with better baselines, and would be happy to include them (e.g. your suggestion of "simple memory-augmented models on top of word embeddings") in the camera-ready version of the paper.
> > >
> > > And yes, probing more profound or wide-ranging knowledge than what we did would be interesting. There are myriad dimensions of knowledge to consider, and we feel that it's important to identify a language model's boundaries in many of them. We started here as a nod to classic tasks from the early days of AI research, and determined important dimensions for generalizing to harder tasks. These include rudimentary ones like tracking an increasing number of straightforward properties about an environment (as with cardinality generalization) or as subtle as the commonness versus the concreteness of the words involved (as with object generalization). The model clearly hasn't mastered these dimensions, and so what we have shown isn't trivial, in our opinion, and reinforces our belief that this work as it stands is valuable to understanding the capacities of the model, no matter how primitive or sophisticated they might seem ex ante. We hope you agree on these points.

---

> > > > ### Comment · Reviewer_gnNv · 2021-08-25
> > > > **Other forms of generalization do not seem clearly separated from basic linguistic understanding**
> > > >
> > > > ```The axes of generalization we explore are useful not just to see whether the model grasps the task by substituting similar words (which we show with part-of-speech generalization, and could be an ability also possessed by embeddings), but also to understand the limits of cardinality generalization in terms of number of objects, containers, steps, and maps tracked, and this is a type of reasoning ability well-detached from learning word-classes. ```
> > > >
> > > >  I agree that these concepts are separate from the question of word classes, but in these games they are built upon them (or finer-grained classes maybe in navigation, to understand what terms are rooms perhaps).  How can you separate models having inductive biases that support reason-based generalizations from simply knowing word-classes if you aren't comparing to a model that one might believe only knows the latter?  Of course the reasoning ability of doing ~8 room navigation tasks from ~7k examples with exact string prediction strikes me as cool.  It wasn't so long ago that I remember LSTMs failing on simpler reversal tasks.  But there's the rub -- as a reader of this work and not a researcher working on these models and similar datasets, I have no point of reference for understanding the difficulty of this sort of task (at least, with 2021-grade tech).  If my only point of reference is given in the paper in the form of a comparison to a model that doesn't begin with any understanding of what a word is, then I would expect a gigantic gap between these models, and it doesn't help me disentangle what types of information from the pre-trained model are helpful for the task, and I think whether or not these go beyond word-classes is not clear.
> > > >
> > > > ```but we thought and think that there is a lot more reason to suspect that large language models would  possess them than that word embeddings would, and for that reason we chose this class of models to base our paper on. We're not aware of any work showing word embeddings to be capable of the things we show in this paper.```
> > > >
> > > > I really agree, but in this context of scientific literature, isn't it reasonable and necessary to see those experiments to understand which types of generalization properties are specific to these types of pre-trained model architectures, and to models trained on various sizes?  In short it would be good to understand that these models offer something unique, and turn some attention to the two ways in which they differ from a typical word2vec setup (model + data size) to understand how they arise.  And only then does it seem to isolate the topic of interest.  I apologize for harping on this idea of word2vec-style baselines, but I have difficulty find a clear novel lesson from these experiments without comparisons of that kind.
> > > >
> > > > Regarding concreteness, I also am not aware of any work showing word embeddings capable of capturing this, but when you consider that the more concrete words should (by virtue of being capable of being interacted with) appear in the contexts of similar action/sensing verbs, it seems likely they would share a similar space in some dimensions.  That is my guess anyway.  I know I've seen similar embedding models improve performance on semantic role labeling beyond using gold postags, and that would imply better performance in subcategorization framing, and that would imply some of these these sorts of interactions probably pop up to some useable extent (or in the lingo of this submission, exist as inductive biases in the representations of semi-supervised embedding models prior to large pre-trained transformer models).

---

### Official Review · Reviewer_KcKG · 2021-07-17

**Rating:** 6
**Confidence:** 5

**Summary:**

This paper takes a closer look at the ability of language models to perform reasoning based tasks such as tracking states of entities and answering navigation based questions. The main question is: Can these models generalize outside of the training distribution, thus exhibiting the ability to learn underlying rules instead of learning superficial correlations that only work on in-distribution data. This question is studied through several different generalization splits. From results, we see some evidence that pre-trained models are able to generalize outside of the training distribution though in some cases, it’s inconclusive, (and I elaborate on why below).

**Main Review:**

Originality: While I think that the exact settings in this work are somewhat original, Clark et al. 2020 (RuleTaker) also  consider reasoning abilities of transformers on synthetic tasks and look at various generalization abilities. Similarly, Banerjee et al. 2020 also look at how well transformers can be trained to do various reasoning tasks in blocks worlds. In the authors’ opinion, what are some major differences between these works and this paper?

Experiments: While I really like this direction, I think some of the experiments are inconclusive in determining whether the model has truly learnt the underlying rule.
- For example, in the containers experiment the model just needs to track the state of 2 containers (which can be easily detected based on the language context) to correctly answer. So, even if the model can generalize to more objects and containers, it is not really impressive since the model can learn to ignore all other containers except the two containers whose states were altered. Indeed, if we look at the navigation results, we see poorer extrapolation since the task is harder.
- In Section 4.4, we cannot immediately conclude that the model is leveraging previous knowledge. The model could just as well be learning superficial features from the container and navigation tasks that help on the hard object tasks. One way to check for this confounder would be to just do some steps of pre-training on the Nav and Cont tasks and see if that by itself explains the fast learning on HardObj. If not, we can then conclude that perhaps the model is leveraging previously acquired knowledge to do well on a composition.

- I really liked the experiment in Section 4.5. Although, I wonder if the decrease in performance has a simpler explanation like gibberish words like “sixnqkxb” contain more sub-words and are hence harder to reason / track.

Quality: I think the paper is decently written. Some style suggestions:
- Table-2 could be converted into a barplot (moving numbers into appendix) since it’s really hard to parse so many numbers. And it’s generally good to bold the best numbers for a quick read.
- In Figure-7, it is unclear what the x axis refers to.


**Time Spent Reviewing:**

3

---

> ### Author Response · Authors · 2021-08-10
> **Response to Reviewer KcKG**
>
> We thank the reviewer for pointing out Clark et al. 2020 and Banerjee et al. 2020, two pieces of work that are relevant to ours. One of the primary differences between our work and both of these papers is that we include plenty of potential distractors in our prompts, unlike these two papers which only provide relevant information to the model.
>
> Rules are made explicit in Clark et al. 2020, whereas we attempt to demonstrate that implicit rules have been learned in ours. For example, we don't ever explicitly say that if the dining room is to the north of the living room, you have to go south to get from the former to the latter but you have to go north to get from the latter to the former, or that once you take a strawberry from the drawer, it is no longer in the drawer. Clark would make these rules explicit.
>
> Banerjee et al. 2020 is similar in the sense that there are knowledge triples, but distinct in that their triples come from formal knowledge graphs, and ours are scenarios of initial states of the environment and either the actions taken on them (container task) or instructions given to navigate them (navigation tasks). The two works probe the language model for different kinds of knowledge.
>
> As for the reviewer's doubts as to the conclusiveness of our experiments, we wanted to point out several points:
>
> With respect to the container task, the reviewer mentions that only two of the containers need to be attended to in any given scenario, and thus that the model could be ignoring the rest of the containers. We do not see a problem with this possibility. If the model attends to the right containers and ignores the wrong containers, that should be taken as a strength of the method. The reviewer wonders whether this is overly simple: Wouldn't it be harder to track three, four, or more containers? We agree that it is less impressive to track two changed containers than three; but by the same logic, it's less impressive to track three than four, and this remains true for every incremental number of changed containers tracked until the task is constrained such that every container needs to change.
>
> The most impressive tasks are not the only ones worth studying, and we consider the ability to track two containers to be both worthwhile in and of itself and fundamental to the ability to track a higher number of containers. Arguably, synthetic tasks are built to shed some light on what models can and cannot do in real-world domains, and while it's true that more than two containers might be manipulated in a real-world task, it's also true that just two might be manipulated.
>
> The model hasn't mastered the simpler task, and so it's obviously not trivial. Thus, it's important to explore it, just as it's important to explore the more difficult task eventually. While it would have been ideal to do so in this paper, we exhausted all the space available to us in a NeurIPS submission. We look forward to tackling that task in future work.
>
> We also want to clarify that in section 4.4, we believe we have performed the experiment the reviewer suggests; all models are trained for 3000 total steps. The difference between the models is that the worst-performing model, "HardObj", learns the composite task exclusively for 3000 steps, and the other three models are trained on the other, simpler tasks sequentially for 2000 total steps and then on HardObj for an additional 1000 steps. The order of this learning is indicated by model's name (e.g. Cont-Nav-HardObj is trained for 1000 steps on Cont, then for 1000 steps on Nav, then for 1000 steps on HardObj). Thus, the evidence presented suggests that the composite learners are leveraging knowledge gained in different stages of training to excel at the HardObj task. However, the reviewer might instead be asking how the model performs after being trained exclusively on the Nav and Cont tasks and before it's ever trained on HardObj, in which case we've also included this information. This is the beginning of the accuracy curves for the composite learners at NTrainingSteps=2000 in Figure 5.
> We are also grateful for the reviewer's other thoughts. We would be happy to make the suggested changes in the camera-ready version of the paper (converting Table 2 into a bar chart or at the very least bolding the top scores, along with clarifying the meaning of the x-axis in Figure 7 (it represents checkpoints taken every 100 steps in fine-tuning for 1000 steps)).

---

### Public Comment · ~Nicolas_Gontier1 · 2022-01-18
**Missing discussion on highly relevant works**

I recently learned about this work. I found it very interesting and well done. Congratulations to the authors on the acceptance.
Nevertheless, I would like to point some **very** relevant work published at NeurIPS 2020 that I worked on:
- Measuring Systematic Generalization in Neural Proof Generation with Transformers https://proceedings.neurips.cc/paper/2020/hash/fc84ad56f9f547eb89c72b9bac209312-Abstract.html

Although we mainly focused on models trained from scratch, we also evaluated the interpolation and extrapolation capacity of transformers regarding the number of reasoning steps. We also concluded that "_longer plans require a longer output. The model seems to struggle with having to generate sentences that are longer than any it has seen before._"

It'd be nice if the authors added a discussion comparing our work to theirs.

---

### Decision · Program_Chairs · 2021-09-27

**Decision:**

Accept (Poster)

**Comment:**

This paper has certainly been put through the ringer of the "moving goalposts" of successive conference reviews, and for that I have a lot of sympathy (although would not make a decision on the basis thereof).

This paper presents an in-depth study of the ability of large pre-trained language models to generalise well on symbolic reasoning tasks, i.e. verifying whether some of the knowledge distilled through the pre-training objective(s) presents a useful inductive bias which aids generalisation in such downstream tasks which have a more structured nature than, say, RTE or QA.

The reviewers concur that the paper is well written, and presents interesting results. Given the interest the community has in this class of models, I concur that this is an important and timely contribution. Reviewer VwQ8 (rating 7) writes in strongest support, and in no small amount of detail, clearly seeking to champion the paper for publication. From reading Reviewer hX8A (rating 6) I could not tell why the rating was not higher. Reviewer KcKG (rating 6) is mildly supporting the paper, while presenting some concerns about whether the experiments are conclusive. I would have liked to read their thoughts following the authors' detailed response, but sadly it seems not to have been forthcoming. Finally Reviewer gnNv (rating 3) wrote at significant length regarding their concerns about this paper, and engaged in what I think must be the most detailed discussion I've witnessed during this round of area chairing.

Academically speaking, I find myself agreeing with a lot of objections and concerns Reviewer gnNv raises. The discussions with the authors here makes for a fascinating read, and I am not clear where the reviewer landed in terms of revising their assessment. I think the fairest way to view the outcome of this conversation is that there are fundamental difficulties with evaluating the sort of question the paper seeks to address, and in a sense the concerns levelled here are a statement about these difficulties more than they are a statement about the appropriateness the authors took, and the rigour thereof. Perhaps the paper is worth publishing on the grounds of its ability to generate such discussion alone.

Taking these reviews holistically, I have formed the strong opinion the paper should be accepted. It generates debate and controversy, but for the right reasons, and attempt to bring further light to the capabilities (and limits) of large pretrained language models. I would hope that the authors could, in as much detail as possible, incorporate their response to reviewers KcKG and gnNv in the final draft of the paper, so that the readers may benefit from these conversations (albeit indirectly).